# Development of Clay-Composite Plasters Integrating Industrial Waste

**DOI:** 10.3390/ma16144903

**Published:** 2023-07-09

**Authors:** Andreea Hegyi, Cristian Petcu, Adrian Alexandru Ciobanu, Gabriela Calatan, Aurelia Bradu

**Affiliations:** 1NIRD URBAN-INCERC Cluj-Napoca Branch, 117 Calea Florești, 400524 Cluj-Napoca, Romania; andreea.hegyi@incerc-cluj.ro; 2Faculty of Materials and Environmental Engineering, Technical University of Cluj-Napoca, 103-105 Muncii Boulevard, 400641 Cluj-Napoca, Romania; 3NIRD URBAN-INCERC Bucharest, 266 Șoseaua Pantelimon, 021652 Bucharest, Romania; 4NIRD URBAN-INCERC Iaşi Branch, 6 Anton Şesan Street, 700048 Iaşi, Romania; bradu_aurelia@yahoo.com; 5Office for Pedological and Agrochemical Studies Cluj-Napoca, 1 Fagului Str., 400483 Cluj-Napoca, Romania; gabi_kavida@yahoo.com

**Keywords:** clay plasters, drying shrinkage, plaster characterization, physical and mechanical properties, sustainable building materials

## Abstract

This research investigates the feasibility of developing clay composites using natural materials and incorporating waste by-products suitable for plastering diverse support structures. The study identified a versatile composition suitable for a wide range of support materials and explored the potential of revaluing industrial waste and by-products by reintegrating them into the Circular Economy. The experimental investigation outlines the process of evaluating the influence of different raw materials on the performance of the clay composite. The findings confirm that using limestone sludge and fly ash as additives to clay contributes to reducing axial shrinkage and increasing mechanical strengths, respectively. The optimal percentage of additives for the clay used are identified and provided. Using hydraulic lime as a partial substitute for clay reduces the apparent density of dried clay composites, axial shrinkage, and fissures formation while improving adhesion to the substrate. Introducing dextrin into this mix increases the apparent density of the hardened plaster while keeping axial shrinkage below the maximum threshold indicated by the literature. Mechanical strengths improved, and better compatibility in terms of adhesion to the support was achieved, with composition S3 presenting the best results and a smooth, fissure-free plastered surface after drying.

## 1. Introduction

Nowadays, there is a global interest in identifying sustainable solutions that are as environmentally friendly as possible for the construction industry. The highly polluting impact of the cement industry—currently the primary raw material used for developing and maintaining the built environment—is well known. Therefore, a series of research studies have turned to innovation based on traditional “heritage” using vernacular materials [1]. The results of this research, besides offering the possibility of preserving traditional identity, support the documented development of construction made from raw-clay-based materials in combination with various additives (mineral, plant, animal, derived from waste or industrial by-products). In addition to the great advantage of reduced environmental impact, buildings made from raw-clay-based materials have several benefits in terms of indoor air quality and, implicitly, the health of the population: water vapor permeability, the ability to regulate indoor air humidity, and high storage/heat release capacity, thus contributing to thermal comfort, increasing indoor air quality and energy efficiency. However, the difficulties these materials present are primarily in terms of mechanical strength and resistance to the action of climatic factors being lower compared to concrete buildings, as well as a reduced degree of compatibility with classic finishing materials available on the market [2,3,4,5,6,7,8,9,10,11,12,13]. Consequently, there is a need to develop plaster and finish materials that are compatible with the primary materials used for traditional construction (natural stone, burnt ceramic brick, and masonry elements based on raw clay, wood, and wood-based or other lignocellulosic materials).

Indoor finishing materials play a significant role in shaping the indoor climate due to their moisture-buffering capabilities. These capabilities arise from the sorption and diffusion properties of the materials, which help maintain a balanced indoor environment [2,3,4,5]. The inherent thermal properties of clay building elements and finishes contribute to their capacity to effectively regulate indoor temperature. This is particularly beneficial when faced with significant daily temperature fluctuations, as highlighted by [6]. Their substantial thermal mass facilitates the absorption and storage of heat throughout daylight hours, subsequently releasing it during cooler periods. This natural temperature modulation fosters a comfortable interior ambiance and impacts the building energy efficiency [7,8,9,10,11,12]. Additionally, clay’s permeability to water vapors enables earthen walls to function as a moisture buffer, adeptly managing humidity levels. Minke shows that, when in equilibrium with saturated moisture air, clay blocks can absorb eight times more moisture than burnt bricks [13]. By absorbing excess water from the indoor air, then gradually releasing it when conditions are dry, clay walls maintain a balanced indoor environment, further enhancing their appeal in contemporary construction. In regions with naturally lower air humidity, such as northern climates, these aspects become particularly important. McGregor [14] shows that the nature of clay minerals has a significant impact on moisture capacity. Moisture storage capacity is affected by the clay particles’ variable surface charge and size. Incorporating finer and more active clay minerals, like montmorillonite, can substantially boost moisture capacity. However, this may also introduce secondary effects, such as increased swelling and shrinkage when moisture levels change. Addressing the issue of increasing water permeability, researchers have discovered innovative methods to enhance the moisture buffering capabilities of clay plasters by incorporating cellulose from waste paper [15]. However, in a milder climate, clay alone may be sufficient to provide humidity-buffering properties. In a newly constructed German house, with both interior and exterior walls made of earth, measurements taken over an eight-year period revealed that the relative humidity inside the home remained consistently close to 50% all year round [13]. This natural regulation of temperature and humidity reduces the need for artificial heating and cooling systems and is beneficial in reducing the risk of respiratory illnesses, allergies, and many other issues.

The specialized literature showcases numerous studies that highlight the durability of buildings constructed using clay and indigenous techniques in diverse geographic and climatic conditions. These studies emphasize the adaptability and resilience of clay-based construction, which has been used for centuries across different cultures and environments [16,17,18,19,20]. The oldest example of using mudbricks in Europe is in Heuneburg, Germany, dating back to the 6th century BCE. Writings by Pliny the Elder indicate the construction of rammed earth forts in Spain at the end of the 100s BCE [20]. Bugini et al. [21] examines a clay plaster consisting of illite, chlorite, kaolinite, and fine quartz, which is believed to be the first-ever identified example in Roman Lombardy. This unique sample was discovered at the Santa Maria alla Porta site, located in the area of Milan’s Imperial Palace, dating back to the first century CE.

In the past couple of decades, houses with earthen building elements have gained significant popularity and recognition. This surge in interest can be attributed to several key factors that make clay houses an appealing choice. However, an essential characteristic is that using natural, non-toxic materials in the building process significantly reduces the risk of indoor air pollution and other health hazards, making clay homes a safer and eco-friendlier option [22]. One important parameter that characterizes the indoor air quality is ozone (O_3_). It readily reacts with various indoor materials and compounds in indoor air, leading to lower indoor ozone concentrations when outdoor air is the primary source. However, the byproducts of indoor ozone reactions can be irritating or harmful to building occupants. Darling et al. [23] suggest that clay-based coatings could serve as efficient passive removal materials, exhibiting relatively low emission rates of by-products that diminish quickly within a two-month period. Another study [24], assessing clay plaster as a passive removal material (PRM) for improving air quality, found that clay plaster improved the perceived air quality and considerably decreased aldehyde concentrations. However, there are relatively few data in the literature on the characteristics of clay plasters [25].

Clay houses have become increasingly popular and well-regarded, with more and more specialists showing interest in this type of construction [26,27]. The results of studies conducted on the possibility of using natural materials in construction are presented in the specialized literature, but these results, in addition to their many benefits, also reveal some drawbacks of clay that are worth investigating for potential improvements [13,28,29,30]. These drawbacks include a high risk of fissuring during the drying process for clay plaster due to significant axial shrinkage and a high sensitivity to water. To be used in the manufacture of construction materials, clayey soil must contain at least 15–16% clay to achieve the necessary plasticity and workability for labor [29,30]. To avoid the appearance of fissures, a linear shrinkage between 3 and 12% for soft mixtures or between 0.4 and 2% for drier mixtures is acceptable [13,28,29,30,31].

Given that there is limited research on the addition of waste materials to clay composites, and considering that in Romania, 80% of such waste is stored in vast areas set up in nature [32,33], constituting a potential environmental hazard, this study aims to valorize these wastes as additives in clay mixtures. Composites based on clay with added ash and sludge have been studied for the production of masonry elements [34,35], and optimal recipes with good results have been developed, thus continuing the study of composites based on clay with added ash and sludge for plastering. The study continues by examining clay mixtures with added ash and sludge for plastering applications.

Clay plasters are known for their eco-friendly and sustainable characteristics [3,13,22,23,24,36] but also have certain drawbacks [13,37]. One of the main concerns is the shrinkage during the drying process. As the water content in the plaster decreases, the material contracts, which can lead to the formation of fissures or cracks. This affects the plaster’s appearance and compromises its ability to adhere properly to the substrate, potentially reducing the structural integrity of the overall construction [13,36,37]. Furthermore, the compatibility of clay plasters with various substrates can be an issue. In construction projects, a wide range of materials may be used as substrates, including wood, brick, stone, and even modern materials like concrete or oriented strand boards (OSB). To simplify the application process and increase the adoption of clay plasters, it is crucial to develop composites that can bond effectively with a diverse array of substrates. This may involve the incorporation of additional binders, fibers, and other additives that can enhance adhesion without compromising the plaster’s environmental profile.

The study focused on examining the potential for developing clay compositions suitable for plastering surfaces constructed from various support materials. The research highlights the impact of different raw materials on the overall performance and characteristics of the clay composite. As a result, the originality and value of the experimental research conducted lie in the following:-Investigating the realm of eco-friendly materials with negligible environmental impact;-Assessing the impact of additional or substitute raw materials on the performance of clay composites;-The objective was to identify a composition that could be widely used across multiple support materials. This would eliminate the need for adapting the plaster composition every time the support layer changes. Ultimately, this would create a favorable framework for achieving a homogeneous and uniform appearance and reduce any delays in the plastering process.-The research aimed to identify ways to reintroduce and revalue industrial waste and by-products. This would help create a favorable framework for implementing the principles of the Circular Economy. By finding new uses for these materials, the research aimed to reduce waste and promote sustainability.

## 2. Materials and Methods

To achieve a low-embedded energy material, it is essential to prioritize the use of locally sourced raw materials. For preparing clay-based composite materials intended for plastering the surfaces of structures, the research focused on locally produced materials, such: clay extracted directly from the soil in Valea Draganului, Cluj Napoca, Romania (46°54′10″ N 22°49′53″ E); fly ash generated by the Mintia power-plant, Romania (a by-product of Romania’s third largest power-plant); limestone sludge, a by-product of limestone processing industry in Hunedoara, Romania; hydraulic lime with the commercial name RÖFIX NHL5; dextrin purchased from a local producer in Cluj Napoca, Romania; and sodium chloride, NaCl, with 90% purity.

Clay is a fine-grained sedimentary rock composed of a blend of silica and fragments of materials such as quartz and mica [38] (X-ray diffraction (XRD) using a high-resolution Brucker D8 diffractometer). The particular clay used in this study, considered to be mostly of montmorillonite type, from the point of view of the particle size distribution, consists of approximately 38.15% clay minerals, 41.93% sand, and 19.92% silt, determined by the sieving and sedimentation method according to the Romanian standard STAS 1913-5 [39]. Based on its size distribution, it is classified as clay loam (Figure 1). It was characterized in terms of oxide composition by XRF analysis and the relevant data are provided in Table 1. From a mineralogical perspective (Figure 2), experimental tests indicated the presence of predominant minerals: quartz 66%, muscovite 32.4%, and vermiculite 1.6%. The tests were carried out in the research laboratories of the Technical University of Cluj-Napoca (XRD analysis) and the Babeș-Bolyai University of Cluj-Napoca (XRF analysis). According to the specialized literature [13,40,41], the mineralogical composition of this raw material indicates that it is appropriate for creating compositions that can be used in construction.

The fly ash underwent a comprehensive analysis to determine its properties and potential applications. Its chemical characteristics were assessed by examining the oxide composition (as shown in Table 2), revealing the percentages of various oxides present in the ash. In addition, the fly ash’s physical properties, such as fineness and bulk density, were evaluated and documented in Table 3. The fineness was determined by dry-sieving using a HELOS RODOS/L, R5 (dry dispersion in the free aerosol jet for laser diffraction), and the bulk density was determined as a ratio between mass and volume using the pycnometer method.

The limestone sludge underwent experimental analysis to determine its properties, focusing on particle size distribution and bulk density, which are presented in Table 4. This characterization provides insights into the sludge’s potential applications and performance in various settings, particularly when used as an additive in construction materials.

The hydraulic lime, introduced into the composite as a substitute for part of the clay, was characterized by experimentally determining the bulk density and compressive strength (Table 5).

Dextrin are constituted from water-soluble amorphous carbohydrates with a small molecule, obtained from the hydrolysis reaction of starch or glycogen, with the chemical formula (C_6_H_10_O_5_)_n_. Dextrin is available as a white, pure powder, soluble in water and poorly soluble in alcohol. When mixed with water, they form a very sticky syrup, which can be used as a substitute for gum arabic and gum tragacanth. Dextrin was characterized by experimentally determining its bulk density, 800 kg/m^3^.

The water used for preparation was dosed so that the sum of the liquid raw material quantities was constant in relation to the sum of the dry material quantities, specifically, 33% by mass, ensuring a consistency of the fresh mixture of 95 ± 5 mm.

### 2.1. Preliminary Experimental Composites

In order to identify and analyze the performance of clay compositions intended for plastering work, a series of mixtures made from three or more of the previously presented raw materials were prepared under laboratory conditions. The compositional design was based on preliminary research regarding the influence of different raw materials on the performance of clay composites and considered a series of guidelines:-Fly ash was considered a potential raw material due to its pozzolanic performance [32,42,43,44,45,46,47,48,49,50,51,52,53,54]. Preliminary experimental research analyzed the axial shrinkage of four clay composites prepared with fly ash used as an additive to a constant amount of clay, specifically 15 g, 20 g, 25 g, and 30 g of limestone sludge added to 50 g of clay, according to Table 6.-Limestone sludge was selected based on preliminary experimental tests and literature reports that indicate beneficial effects [13,41,55,56,57] due to the additional calcium oxide contribution in reducing the volume variations of the composite. Thus, preliminary experimental research analyzed the axial shrinkage of four clay composites prepared with limestone sludge used as an additive to a constant amount of clay, precisely 15 g, 20 g, 25 g, and 30 g of limestone sludge added to 50 g of clay, according to Table 7.-NaCl was identified in preliminary tests, and in accordance with the specifications of the specialized literature [13,41,58,59,60,61], as a beneficial additive in clay-based composites. It helps in reducing axial shrinkage and minimizes the risk of cracking by moderating and standardizing the drying process. In clay composites, NaCl was introduced as an 8% saline solution prepared by dissolving the salt in potable water.-Hydraulic lime was chosen based on initial experimental tests and numerous literature findings that highlight its positive impact in reducing drying shrinkage and the likelihood of fissuring, due to the added calcium oxide content [62,63,64,65,66,67,68,69,70,71]. In this instance, preliminary experimental research examined the effects of partially substituting clay with hydraulic lime in compositions consisting of clay, fly ash, and limestone sludge. The study focused on evaluating the risk of fissuring, adhesion to clay-based support, axial shrinkage, apparent density, and mechanical resistance in four clay composites, as outlined in Table 8. In all the mixtures, sodium chloride was incorporated as an 8% mass concentration aqueous solution. The amounts of fly ash and limestone sludge were kept constant. The ratios were determined based on prior experimental studies, keeping a constant mass proportion of additive materials (limestone sludge/fly ash) to clay. In instances where clay was partially replaced by hydraulic lime, a constant mass proportion of additive materials (limestone sludge/fly ash) to the combination of (clay + hydraulic lime) was maintained.-Dextrin was selected based on literature reports indicating beneficial effects of similar organic additives [3,37,72,73,74], which include improved paste workability, decreased drying rate, increased uniformity in the drying process, and, ultimately, a reduction in fissures within the dried plaster layer.

### 2.2. Clay-Based Compositions Developed in Accordance with the Preliminary Research Findings

Following the preliminary experimental research results, the aim was to determine the impact of dextrin on the performance of clay composites for plastering surfaces in eco-traditional buildings. As a result, five different composites were developed, as shown in Table 9.

The chosen substrates for applying clay composite plasters were selected from commonly used construction materials in eco-traditional wall buildings. These include ceramic bricks, limestone, oriented strand boards (OSB), unfired clay masonry units, and masonry elements made from a composite material consisting of clay and cereal straw.

### 2.3. Experimental Testing Methods

For preliminary clay composites PA1–PA5, PS1–PS5, PV1–PV5, and dextrin-containing clay composites S1–S5, prismatic samples of 40 × 40 × 160 mm were produced by casting in metal molds. After maintaining for 24 h in constant laboratory conditions of 23 ± 2 °C and 65 ± 1% relative humidity, the samples were removed from the molds and stored for 40 days for curing/drying.

For the clay composites PV1–PV5 and S1–S5, which underwent experimental testing for adherence to the substrate, the process involved preparing flat samples using different substrates. These substrates included ceramic bricks, limestone, OSB boards, unfired clay masonry units, and masonry elements made of composite material based on clay and cereal straw. A 3–5 mm thick layer of the clay composite material was applied to the previously mentioned substrates. The surfaces of the substrate were treated with bone glue aqueous solution before applying the clay composites. For this purpose, an 8% concentration bone glue solution was prepared by dissolving bone glue in warm water. The application of the clay composites was carried out only after the drying of the support surfaces treated with the bone glue solution.

All samples were stored for 40 days until reaching constant mass. The constant mass of the sample was regarded as an indication of both material maturity and dryness. The samples were maintained under constant laboratory conditions of 23 ± 2 °C and 65 ± 1% relative humidity and were then experimentally tested under the same environmental conditions to analyze the following parameters:-identification of fissures through visual analysis;-axial shrinkage after 40 days of drying, following the testing method established by Romanian standard STAS 2634 [75], as a percentage reduction of the length of the specimen when it reaches the equilibrium humidity (40 days), in relation to the initial length recorded at de-molding;-apparent density in hardened and dried state, 40 days after casting, following the testing method established by European standard EN 1015-10 [76], as a ratio between the mass and the volume of the specimen;-mechanical strengths of the dried state, 40 days after casting, following the testing method established by European standard EN 1015-11 [77], using an automatic press. In order to determine the flexural strength, the concentrated load method was used halfway between the supports, positioned at a distance of 100 mm from each other, each at a distance of 20 mm from the ends of the prismatic specimen. Later, on the two halves of the prism resulting from the tensile testing by bending, investigation was carried out to determine the compressive strength as a ratio between the maximum load recorded at the time of breaking and the surface of the plates through which the compressive stress was applied (40 × 40 mm).-adherence to substrate in the dried state, 40 days after casting, following the testing method established by European standard EN 1015-12 [78], by the pulling method, using an Elcometer pull-off device.

To ensure repeatability of the experimental tests, a minimum of 3 samples were prepared for each case. The reported values represent the arithmetic mean of individual values.

## 3. Results and Discussions

### 3.1. Preliminary Experimental Research Results

The preliminary experimental research revealed aspects that were later incorporated into the design of the tested clay compositions to determine the optimal variant for creating plaster material:-Using limestone slurry as an additive to clay, as in Figure 3, resulted in a 1.63–2.54% reduction in axial shrinkage compared to the control clay sample (without limestone slurry). The most suitable limestone slurry to clay ratio, which produced the desired effect of reducing axial shrinkage, was determined to be 1:2 parts by mass (sample code PS4).-Incorporating fly ash as an additive to clay increased mechanical strengths relative to the control clay sample (without limestone slurry). This aligns with the findings of other experimental studies in the field [79]. While compressive strength increases with higher levels of fly ash (as shown in Figure 4), flexural tensile strength was adversely affected when the ratio of fly ash to clay exceeded a certain percent (as demonstrated in Figure 5). Aggregating this information, the optimal ratio of fly ash to clay from the point of view of mechanical strengths was established as 1:2 parts by mass (sample code PA4).-Using hydraulic lime as a partial substitute for clay led to a significant reduction in the apparent density of dried clay composites, axial shrinkage, and mechanical strengths (Figure 6, Figure 7, Figure 8 and Figure 9), especially when the amount of clay replaced by hydraulic lime was higher. In this context, it is important to weigh the advantages and disadvantages of using hydraulic lime as a substitute for clay in these composites, which are intended for plastering surfaces of eco-traditional construction. On one hand, the use of hydraulic lime leads to reduced axial shrinkage, which is a positive outcome. On the other hand, it results in decreased mechanical strengths, which is not advantageous. Therefore, the presence or absence of fissures and the adherence of the composites to clay support (clay masonry elements) are considered crucial factors in determining the appropriate quantity of hydraulic lime to be used. Visual evaluation of the samples indicated the absence of fissures for composites PV1–PV3. The PV4 composite detached from the support while drying, making visual evaluation inconclusive in this instance. However, the maximum adherence to support was recorded for composite PV2, in which clay was replaced by hydraulic lime at a ratio of 40% (mass percentages). Therefore, this composite is considered the most appropriate option to further experimental investigation.

### 3.2. Results of Experimental Research for Clay-Based Compositions Developed Following Preliminary Research Results

The experimental results obtained are presented in Figure 10, Figure 11, Figure 12 and Figure 13. Experimental outcomes for clay compositions PV1–PV4 and S1–S5 (Figure 10) showed axial shrinkage below the maximum allowable limit indicated by specialized literature (12%). Therefore, all the analyzed compositions satisfy this requirement from this viewpoint.

Considering that these mixtures are intended for plastering applications, it is important to attempt for reduced shrinkage levels. By doing so, the potential for cracking and fissuring can be minimized, ensuring a more durable and long-lasting plaster performance. Clay compositions PV3 and PV4, which contain a large amount of hydraulic lime, have the lowest axial shrinkage compared to the other samples. Moreover, analyzing the axial shrinkage recorded for clay compositions PV1–PV4 (compositions that do not contain dextrin), a decreasing trend of this indicator is observed as the amount of lime increases. Therefore, the beneficial effect of lime in reducing the axial shrinkage of the composite is confirmed. On the other hand, the introduction of dextrin into the composite matrix results in increased axial shrinkage. The values are within a range comparable to the axial shrinkage of sample PV1, which contains neither hydraulic lime nor dextrin. However, these compositions (S1–S5) possess a lower lime content compared to the compositions that demonstrated the best performance regarding axial shrinkage. When comparing the experimental results of samples S1–S5 to the performance of sample P2 (composition without dextrin, identified as the most suitable for designing composites S1–S5), there was a 37–44% increase in axial shrinkage. However, since these values fall within the limit specified by specialized literature (max. 12%), this increase was not considered a drawback for the development of clay composites with dextrin intended for plastering.

According to specialized literature, density values within the range of 1600 kg/m^3^ and 2000 kg/m^3^ provide the advantage of good thermal resistance for the material [2,8,13,14,19]. However, these indications from the literature mainly consider the behavior of masonry elements intended for constructing walls made of clay composites. In this case, the intended use of the analyzed clay composites is plastering. Therefore, although thermal resistance characteristics are not negligible, they are not as important a parameter in performance evaluation. Consequently, it was determined that several factors are more crucial for evaluating composite performance. These include a smooth surface free from fissuring risks, strong adherence to the supporting material, vapor permeability, and resistance to environmental agents.

The experimental results recorded for the density of the analyzed clay composites (Figure 11) indicated several interesting aspects. It can be observed that using hydraulic lime as a substitute for clay in clay composites PV2–PV4 leads to a reduction in apparent density by 8%, 20%, and 25%, respectively, compared to the apparent density of the clay composite prepared without hydraulic lime (PV1). However, this decreasing trend in apparent density is not maintained in the case of clay composites prepared with dextrin (compositions S1–S5). In this case, the parameter followed shows a slight increase, even above the 1600 kg/m^3^ limit indicated by specialized literature as the minimum for good thermal resistance behavior. Thus, in the case of compositions with dextrin, apparent density values ranging from 1610 to 1623 kg/m^3^ were recorded, increasing as the amount of dextrin used was higher. Therefore, it is considered that the use of dextrin in clay composites brings slight benefits, leading to an apparent density within a range of values favorable for good thermal resistance behavior.

Regarding the compressive strength of the analyzed clay composites (Figure 12), two clear trends were observed: on one hand, the partial substitution of clay with hydraulic lime leads to a decrease in compressive strength by approximately 45%, 58%, and 64% (samples PV2–PV4). On the other hand, the introduction of dextrin into the clay composite (samples S1–S5) slightly improves compressive strength, with the improvement increasing as the amount of dextrin used is higher. Comparing the experimental data to sample PV1, which lacks hydraulic lime, reveals a 30–33% reduction in compressive strength for composites made with partial substitution of clay with lime and added dextrin (samples S2–S5). This is seen as a minor improvement in performance when contrasted with the 45–64% decrease observed in samples without dextrin. Still, considering that samples with dextrin also have hydraulic lime, their compressive strength, although improved, does not reach the value recorded for sample PV1, the clay composite made without partially substituting clay with lime. In this case, when comparing the experimental results of samples S1–S5 with the performance of sample P2 (the composition without dextrin identified as the most suitable for designing compositions S1–S5), an increase in compressive strength for composites with a minimum content of 4 g dextrin per 50 g (clay + hydraulic lime) was observed. Examining the compressive strengths of compositions with dextrin (S1–S5) reveals that this additive may have a minor positive impact on the composite’s mechanical performance. However, this beneficial effect is constrained, with a maximum benefit that cannot be surpassed.

On the other hand, analyzing the experimental results in the context of the classification limits indicated by EN 998-1 [80], a specific standard for industrially produced plaster and skimming mortars based on inorganic binders, it can be said that all tested clay composites fall within the minimum limits imposed by this standard (0.4 N/mm^2^—minimum value for class CS I). Furthermore, all tested clay composites could be classified in the second class in terms of compressive strength, CS II, indicated by this standard, except for the clay composite PV1, which would fall into a higher class, namely CS III. This finding is particularly valuable in terms of the applicability of these composite materials. The ease of their in situ use increases when they can be compared to cement-based plaster or skimming mortar products, which are commonly employed in the construction industry.

In terms of flexural tensile strength, as shown in Figure 12, a pattern similar to compressive strength is observed. With the partial substitution of clay with hydraulic lime in samples PV2–PV4, the flexural tensile strength decreases by 18%, 32%, and 47%. This reduction is in comparison to the flexural tensile strength of the composite that does not contain lime (sample PV1). Upon introducing dextrin to the clay composition, there is a visible trend of reduced strength loss as the dextrin content increases. This trend continues up to a certain threshold, as seen in samples S1–S5. However, when the amount of dextrin is further increased, the mechanical performance and flexural tensile strength decline, as observed in samples S4–S5, falling below the maximum value recorded for sample S3. This behavior suggests that the flexural tensile strength can experience slight improvement with the addition of a small amount of dextrin in the clay composite, but the extent of this improvement is limited. This finding aligns with the observation made during the analysis of compressive strength performance. As a result, it can be inferred that while adding dextrin to the clay composition may provide some benefits to the mechanical performance, these positive effects are relatively reduced and limited.

Analyzing the compressive strength and axial shrinkage for the clay composites S1–S5, in relation to their density, it was observed that these parameters show a progression that can be mathematically expressed through functions of density (f(density)), as shown in Figure 13, with a sufficiently good accuracy indicated by the R^2^ factor being greater than 0.9. Both compressive strength and axial shrinkage can be mathematically modeled using polynomial functions. In both cases, the existence of these functions allowing mathematical reproducibility with a high degree of accuracy is considered a useful tool for designing subsequent clay compositions.

Identifying mathematical functions for adhesion to the support in relation to density or compressive strength of clay composites is not as straightforward. In this case, mathematical functions satisfying the R^2^ > 0.9 condition are polynomial functions of degree 3 or higher, indicating reduced accuracy and challenges in modeling the phenomenon. This implies that numerous factors influence the adhesion parameter to the support, and it does not solely depend on the clay composite’s characteristics, as initially anticipated.

The key characteristics of plaster mortars involve their ability to adhere strongly to substrates and resist fissuring. A good plaster mortar should maintain a solid bond with the underlying surface while remaining fissure-free, ensuring durability and a visually appealing finish. These properties are crucial for the performance and longevity of plastered surfaces in construction applications. Regarding the adhesion to the clay composite masonry element substrate, Figure 14 shows an undeniable increase in this parameter with the introduction of dextrin in the composition (S1–S5 vs. PV1–PV4). It is important to note that the compositional design of samples S1–S5 was based on composition PV2. This further highlights the advantages of adding dextrin to the mixture, as it significantly improves the adhesion of the composites to their respective substrates.

When examining the behavior of clay composites that incorporate dextrin (S1–S5), it becomes evident that composition S3 stands out, exhibiting the best performance irrespective of the underlying support layer. Concurrently, the type of support appears to play a role in influencing the adhesion of the clay composite. Different support materials might lead to varying levels of adhesion, which could affect the overall performance and stability of the clay composite when applied.

To enhance adhesion to the support, before the clay composites were applied, all support surfaces were pre-treated with an 8% aqueous solution of bone glue. Despite this treatment, the oriented strand board (OSB) surface seems to pose the greatest challenge when it comes to compatibility with clay-based plaster. This varying behavior of clay plasters in relation to different support types may be due to specific properties of the support surface, such as water absorption capacity and surface roughness. This hypothesis can also be supported by analyzing the presence or absence of fissures that appeared at the end of the drying period on the surface of clay plasters.

A visual analysis of the surface appearance of various clay composites, after drying, is provided, which can be found in Table 10. This analysis offers insights into the texture and appearance of these composites, which can clarify their suitability for different applications and their overall performance in terms of aesthetics and functionality. It can be observed that there are cases of micro-fissuring, fissuring, and cases where fissures or even detachment from the surface are identified (composition S1 on OSB support), depending on the type of composition and the nature of the support, but there are also cases where the dried plaster surface is smooth, without traces of fissuring or other defects. Composition S3 is identified as having the best compatibility with all analyzed supports. Figure 15 shows exemplary images of the samples made by applying clay composites S1–S5 on different support layers.

Examining the adherence performance of clay composites S1–S5 to various supports (as depicted in Figure 14), we find that the results are rather promising when compared with benchmarks found in the literature (Table 11). The minimum recorded value was 0.3 N/mm^2^ and the maximum was 0.6 N/mm^2^, suggesting the suitability of these composites for both finishing and decoration purposes in construction. When cross-reference the obtained data with other studies, as represented in Table 11, it is noticed that both limestone sludge and, more notably, hydraulic lime result in reduced axial contractions. These contractions are within the 12% limit suggested by the literature. The use of hydraulic lime, and particularly fly ash, in clay composites enhances mechanical strength, both in terms of compression and bending. Regarding the density of the clay composite, it is slightly lower than average but still aligns with references from the literature. Given these data, we can conclude that our results are reasonably in line with those from other studies. However, we must take into account two important considerations:-the experimental findings cited in scholarly sources are influenced by various factors. These include the clay’s compositional, oxide and mineralogical characteristics, along with the type and quantity of added materials [13,18,21,26,29,41,67,81]. These characteristics can vary based on the region where the clay is sourced. Hence, any comparison of these results should be general, focusing on the overall trends in physical and mechanical properties, rather than exact numerical similarities;-From the point of view of adherence to the support layer of the composite material based on clay, performance improvements can be achieved. This improvement was notably demonstrated in the later developed composites, namely S1–S5.
materials-16-04903-t011_Table 11Table 11Benchmarking against previous research findings.Bibliographic Reference/Sample CodeAxial Shrinkage [%]Compression Strength [N/mm^2^]Flexural Strength [N/mm^2^]Density [kg/m^3^]Adherence to Support [N/mm^2^]Clay from Dallgow-Döberitz area, Germany [13]15.03.300.631748-Clay from Cluj-Napoca area,Romania [19]-2.40-1960-Four clay types from Toulouse area, France [25]1.5–2.11.30–2.100.49–0.64-0.06–0.8Clay from Albi, France [25]2.51.700.57-0.06Clay from Poland [37] 2.21.34–1.500.49–0.581802–1853
Clay from Oluvil, Sri Lanka [41]-2.8-8.9
-0.15-0.44Clay from Chom Thong, Thailand [82]17.10.32-1050-Clay from Middle Belt, Nigeria [83]25.010.2-1700-Clay from Guimaraes, Portugal [84]-1.50-17480.10Clay from Bath area, UK [85]-2.50
19600.11–0.28Sandy-loam soils from Kôdéni, Burkina Faso [86]3.131.800.571720-Clay used for the developed composites9.204.301.551600-Composites PS2–PS58.97–9.05----Composites PA2–PA5-6.95–7.801.55–1.83--Composites PV1–PV42.70–6.402.1–5.80.9–1.81190–1606.70.0–0.4Composites S1–S5Masonry element made of unfired clay6.3–6.62.6–4.11.2–1.51610–16230.5–0.6Ceramic brick0.5–0.6Masonry element made of clay and cereal straw0.4–0.5Limestone0.4–0.6OSB0.0–0.4

## 4. Conclusions

The experimental findings are remarkable and practical because the research focused on identifying and highlighting the impact of alternative or additive raw materials, such as clay, and by-products from other industries on the performance of clay composites.

The experimental research investigated the potential of using clay composites as plasters for various surfaces commonly found in eco-traditional construction. These surfaces included ceramic brick, limestone, OSB boards, masonry elements made of clay, and masonry elements made of composite material based on clay and cereal straw. Based on this research, the following observations can be made:-The research found that adding limestone sludge to clay helped reduce axial shrinkage. For the best results, the recommended ratio of limestone sludge to clay was 1:2 by mass;-It is demonstrated that adding fly ash to a clay composite resulted in increased mechanical strengths, particularly compressive strength. For the best results, the recommended ratio of fly ash to clay was 1:2 by mass;-Utilizing hydraulic lime as a partial substitute for clay leads to a decrease in the apparent density of dried clay composites, axial shrinkage, and mechanical strengths. This reduction is more significant when a larger amount of clay is replaced by hydraulic lime. Additionally, this partial substitution has a positive impact on minimizing fissures formation and enhancing adhesion to the substrate. As a result of the experimental findings, the preliminary composition PV2 was determined to be the most appropriate for designing and developing dextrin-containing clay composites;-The addition of dextrin to clay composites with fly ash, limestone sludge, and hydraulic lime resulted in a higher apparent density of the hardened mortar. This increase surpassed the minimum threshold of 1600 kg/m^3^, which is considered to provide good thermal resistance according to the specialized literature. Furthermore, clay composites with dextrin exhibited axial shrinkage below the maximum threshold of 12% mentioned in the literature. These composites also showed improved mechanical strengths and better adhesion compatibility to the support. Composition S3 demonstrated the most favorable outcomes, displaying a smooth and fissure-free surface after drying.

The experimental research, on the one hand, demonstrates the possibility of making clay-based plastering mortars even at an industrial level. On the other hand, it underlines the importance of a detailed analysis of the clayey raw material whose characteristics differ depending on the place of extraction. Therefore, each case must adapt and customize the recipe and the production process. These observations and the need for customized design need to be extended to fly ash, whose characteristics differ from one power plant to another.

## Figures and Tables

**Figure 1 materials-16-04903-f001:**
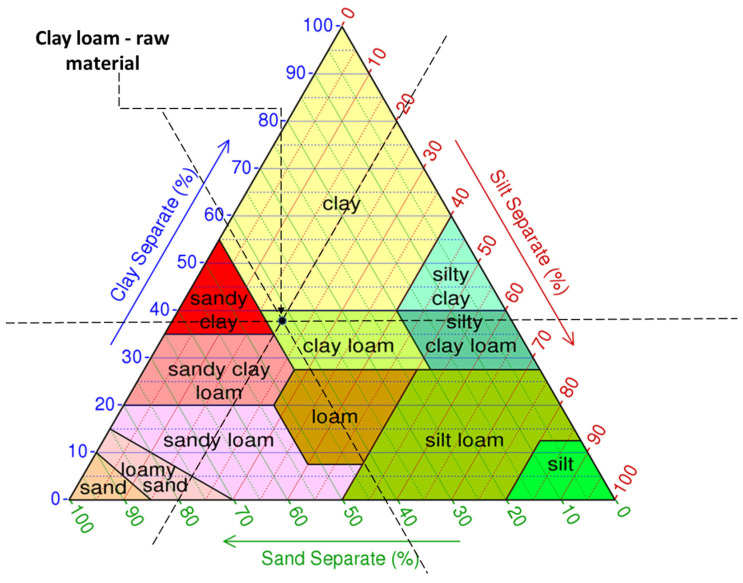
Characterization of the clay used as raw material—material placement in the ternary diagram.

**Figure 2 materials-16-04903-f002:**
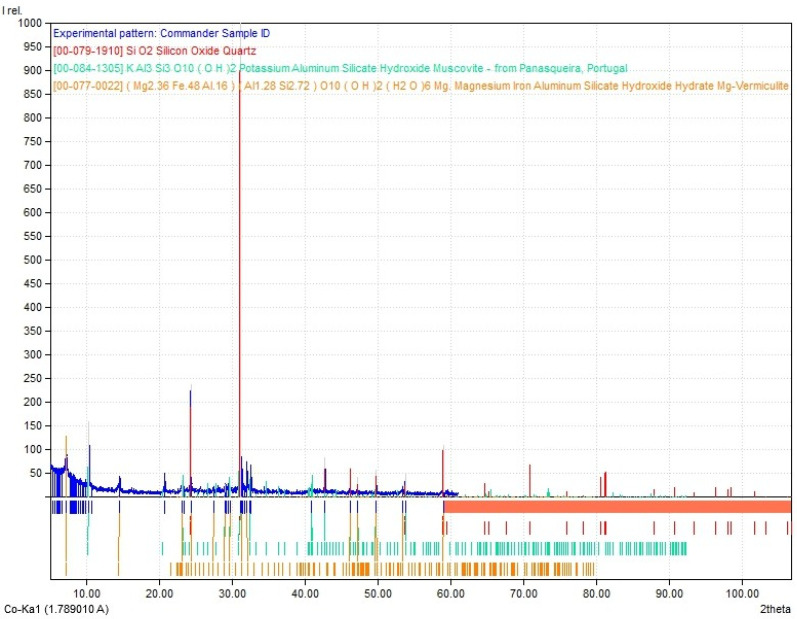
Characterization of the clay used as raw material—mineralogical analysis, determined by XRD analysis (X-ray diffraction).

**Figure 3 materials-16-04903-f003:**
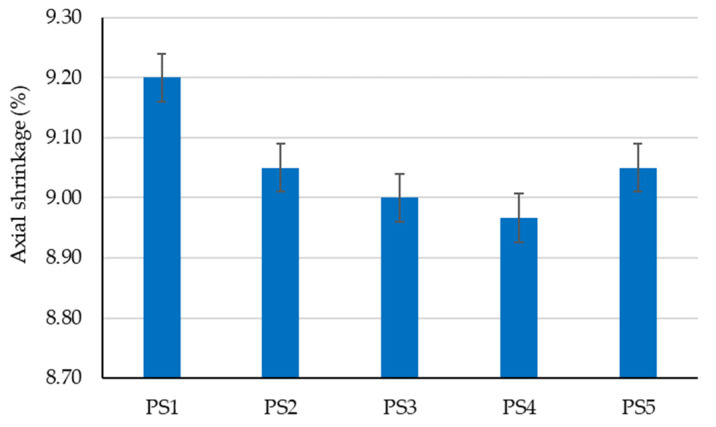
The influence of adding limestone slurry on the axial shrinkage of the clay material.

**Figure 4 materials-16-04903-f004:**
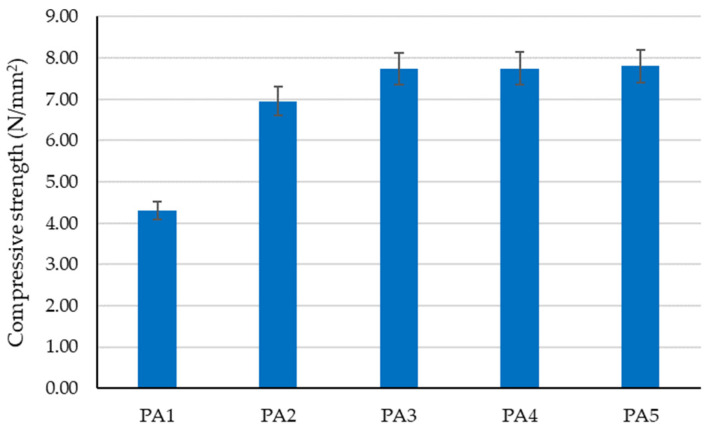
The influence of adding fly ash on the compressive strength of the clay material.

**Figure 5 materials-16-04903-f005:**
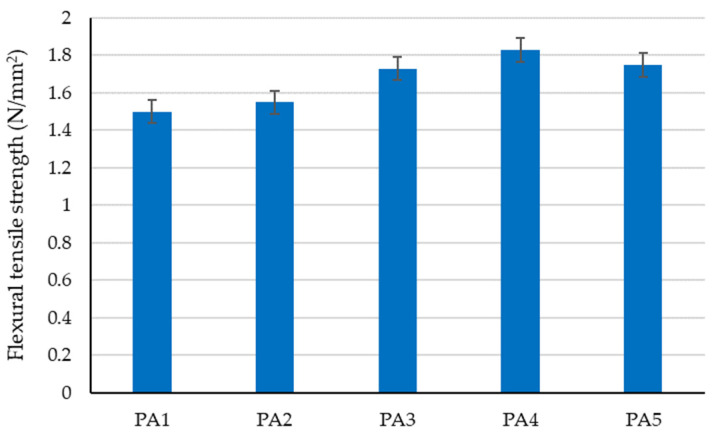
The influence of adding fly ash on the flexural tensile strength of the clay material.

**Figure 6 materials-16-04903-f006:**
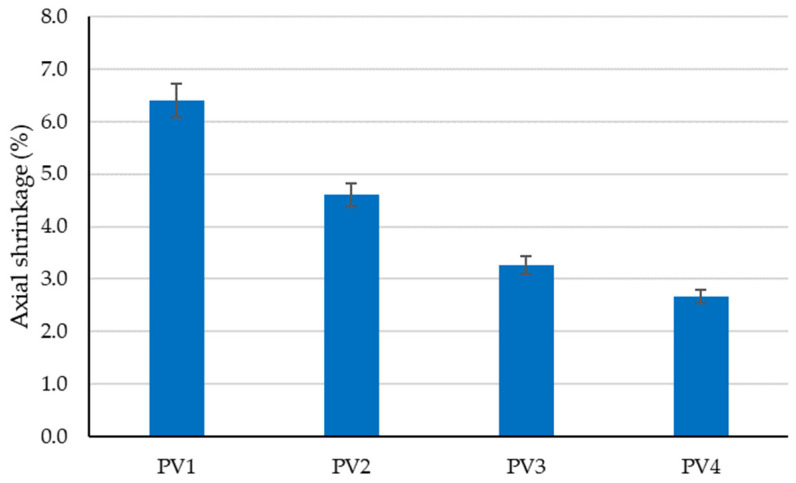
Influence of partial substitution of clay with hydraulic lime on the axial shrinkage of the clay material.

**Figure 7 materials-16-04903-f007:**
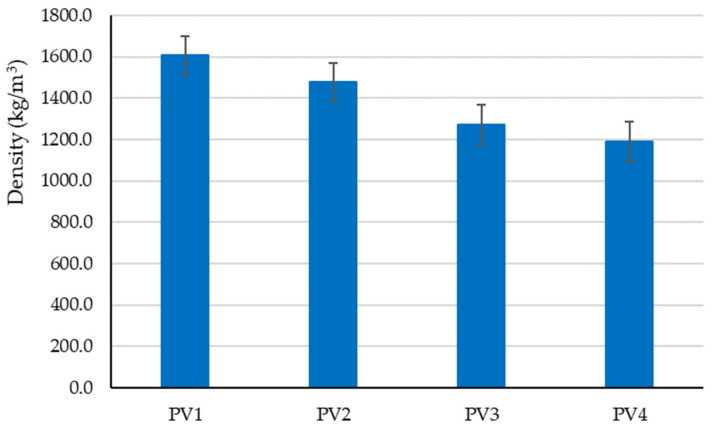
Influence of partial substitution of clay with hydraulic lime on the apparent density of the clay material.

**Figure 8 materials-16-04903-f008:**
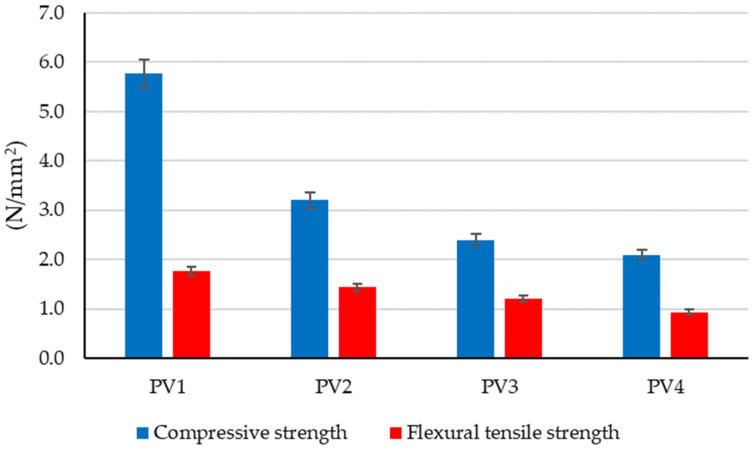
Influence of partial substitution of clay with hydraulic lime on the mechanical strengths of the clay material.

**Figure 9 materials-16-04903-f009:**
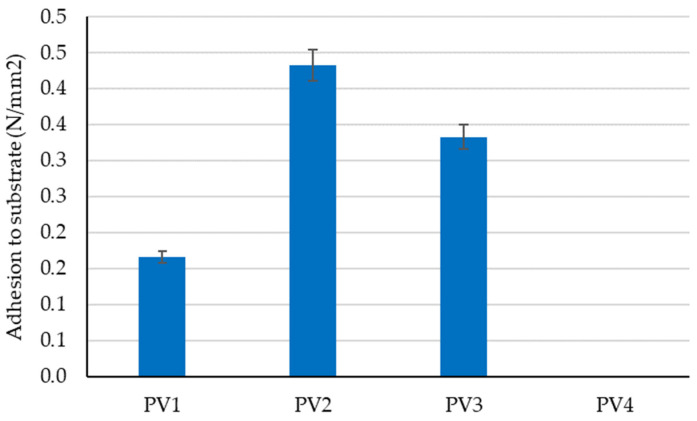
Influence of partial substitution of clay with hydraulic lime on the adhesion of the clay material to a clay-based masonry element support.

**Figure 10 materials-16-04903-f010:**
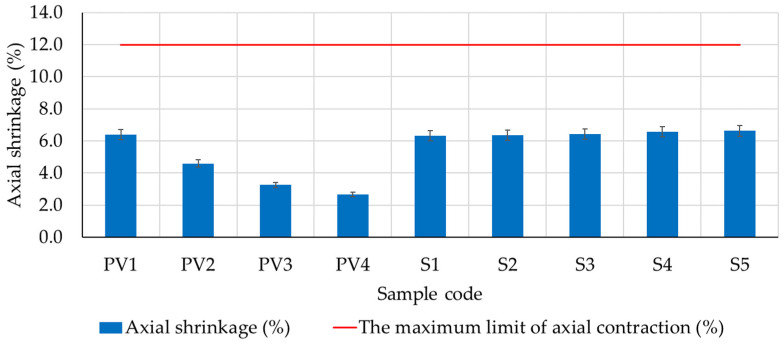
Axial shrinkage of clay mixtures after 40 days of maturation in constant laboratory conditions.

**Figure 11 materials-16-04903-f011:**
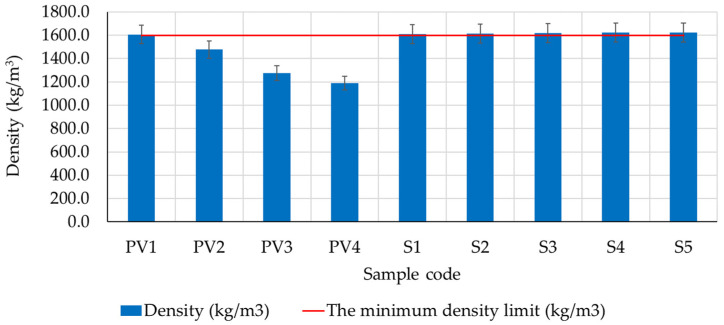
The apparent density of the clay composites after 40 days of maturation under constant laboratory conditions.

**Figure 12 materials-16-04903-f012:**
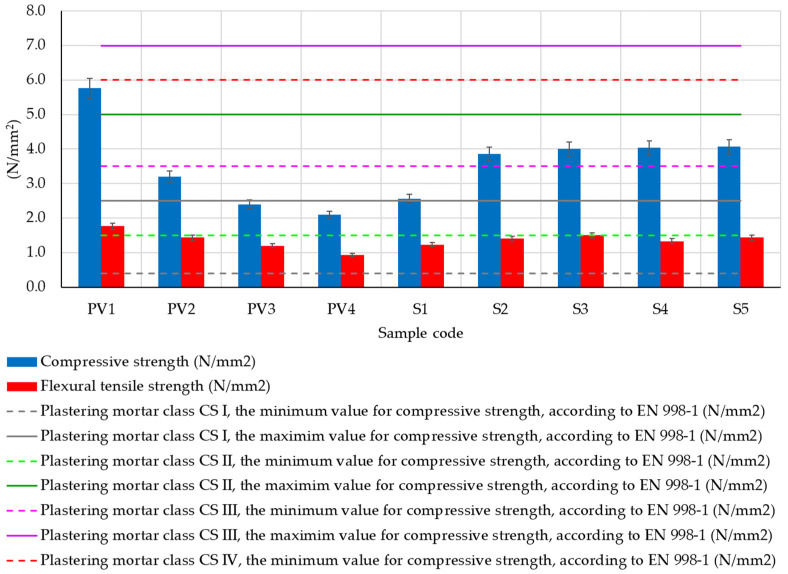
Resistance to compression and flexural extension of clay composites after 40 days of maturation under constant laboratory conditions.

**Figure 13 materials-16-04903-f013:**
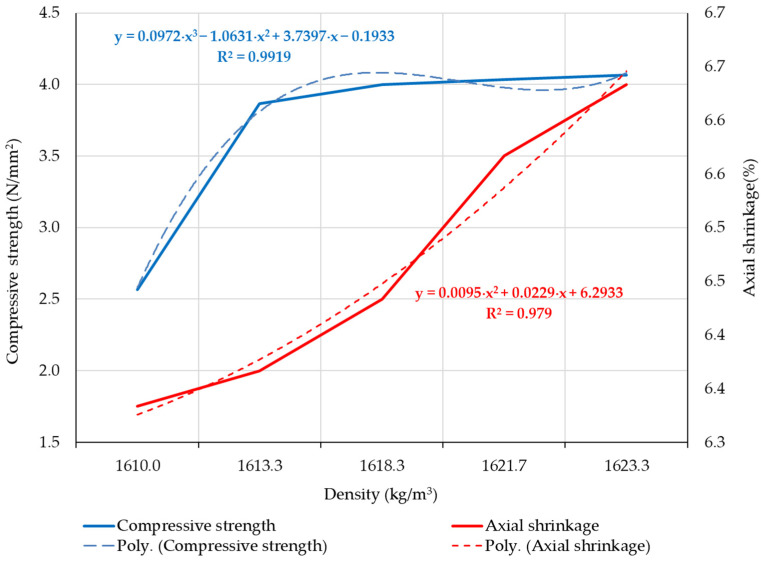
The evolution of compressive strength and axial shrinkage depending on the apparent density of the clay composite.

**Figure 14 materials-16-04903-f014:**
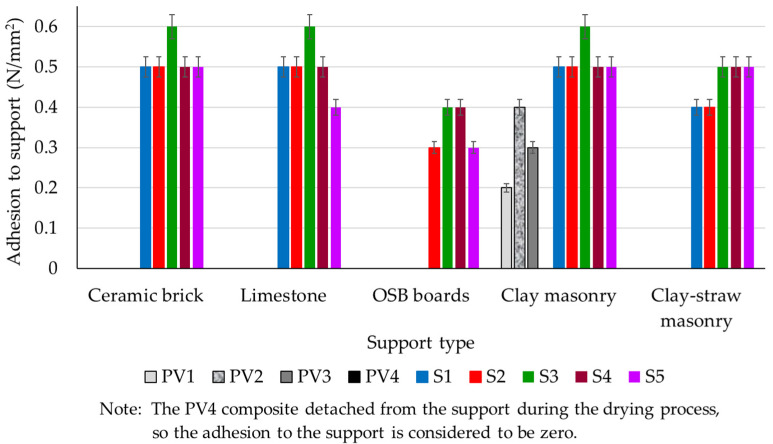
Adhesion to the support of clay mixtures after 40 days of maturation under constant laboratory conditions.

**Figure 15 materials-16-04903-f015:**
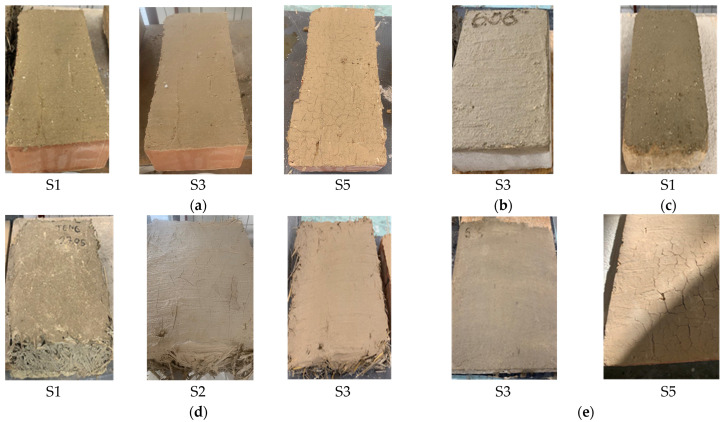
Representative images presenting the application of clay composite S1–S5 on various support materials, illustrating their compatibility and appearance after drying. (**a**) Ceramic brick support; (**b**) limestone support; (**c**) clay support; (**d**) support made of composite material based on clay and cereal straw; (**e**) OSB support.

**Table 1 materials-16-04903-t001:** The oxide composition of the clay, determined by XRF analysis (X-ray fluorescence).

Oxides	SiO_2_	Al_2_O_3_	Fe_2_O_3_	CaO	MgO	K_2_O	Na_2_O	TiO_2_	PC
Content [%]	74.17	12.74	4.38	0.7	1.0	1.43	0.73	0.05	4.78

**Table 2 materials-16-04903-t002:** The oxide composition of the fly ash, determined by XRF analysis (X-ray fluorescence).

Oxides	SiO_2_	Al_2_O_3_	Fe_2_O_3_	CaO	MgO	SO_3_	Na_2_O	K_2_O	P_2_O_5_	TiO_2_	Cr_2_O_3_	Mn_2_O_3_	ZnO	SrO	PC
Content [%]	53.75	26.02	7.91	2.54	1.54	0.35	0.59	2.57	0.12	1.02	0.05	0.09	0.02	0.03	2.11

**Table 3 materials-16-04903-t003:** Physical characteristics of ash samples.

**Fly Ash from Mintia** **Power Plant**	**Fineness R_0.045_**	**Apparent Density (Mg/m^3^)**
	39.20%	1.67

**Table 4 materials-16-04903-t004:** Physical characteristics of the limestone sludge.

Sieve mesh sizes (mm)	0.063	0.125	0.250	0.500	1
Passings through the sieve (%)	69	93	95	99	100
Apparent density (kg/m^3^)	1780				

**Table 5 materials-16-04903-t005:** Physical characteristics of the hydraulic lime.

Characteristics	Apparent Density (kg/m^3^)	Compression Strength (N/mm^2^)
Hydraulic lime	520	5

**Table 6 materials-16-04903-t006:** Composition of the preliminary clay composites experimentally prepared to assess the influence on mechanical resistance of adding fly ash to clay.

Sample Code	PA1	PA2	PA3	PA4	PA5
Clay (g)	50	50	50	50	50
Fly ash (g)	-	15	20	25	30

**Table 7 materials-16-04903-t007:** Composition of the preliminary clay composites experimentally prepared to assess the influence on axial shrinkage of adding limestone sludge to clay.

Sample Code	PS1	PS2	PS3	PS4	PS5
Clay (g)	50	50	50	50	50
Limestone sludge (g)	-	15	20	25	30

**Table 8 materials-16-04903-t008:** Composition of the preliminary clay composites experimentally prepared to assess the influence of the partial substitution of clay with hydraulic lime.

Raw Materials	Clay (g)	Hydraulic Lime (g)	Fly Ash (g)	Limestone Sludge (g)	Saline Solution NaCl, 8% (g)
Clay Composition Code					
PV1	50	0	25	25	17
PV2	30	20	25	25	17
PV3	25	25	25	25	17
PV4	20	30	25	25	17

**Table 9 materials-16-04903-t009:** Composition of the experimental clay composites (percentage composition reported to the total mass of the mixture).

Raw Materials	Clay (g)	Lime (g)	Fly Ash (g)	Limestone Sludge (g)	Saline Solution NaCl, 8% (g)	Dextrin (g)
Clay Composition Code						
S1	30	20	25	25	17	2
S2	30	20	25	25	17	4
S3	30	20	25	25	17	6
S4	30	20	25	25	17	8
S5	30	20	25	25	17	10

**Table 10 materials-16-04903-t010:** Visual analysis of the clay composite layer applied to the support, after drying.

Clay Composite Code	The Appearance of Plaster Layer after Drying				
	Ceramic Brick	Limestone	OSB	Masonry Element Made of Unfired Clay	Masonry Element Made of Clay and Cereal Straw
S1	no fissures	no fissures	detachment	no fissures	fissures
S2	no fissures	no fissures	fissures	no fissures	fissures
S3	smooth appearance, without fissure	smooth appearance, without fissures	smooth appearance, without fissures	smooth appearance, without fissures	no fissures
S4	no fissures	micro-fissures	fissures	micro-fissures	micro-fissures
S5	fissures	fissures	Major fissures, cracks	fissures	deep fissures

## Data Availability

Not applicable.

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
