# Peer review of "Development of Clay-Composite Plasters Integrating Industrial Waste"

_materials, 2023, doi:10.3390/ma16144903_

Round 1
Reviewer 1 Report
Firstly, I deeply thank the authors for their efforts. And I have some comments that t hope to be regarded.
1-The author needs to clarify the novilty of his work.
2- very little characterization technics used.
3- the author should provide the industrial outcomes from this study as they used a worldwide clay as well as waste. So that, they have to highlight the NOVEL industrial benefits in case their work will be applied on a large scale.
4- the introduction is very very long, from my point of view it must be shortened.
Author Response
Dear Reviewer, we deeply appreciate your thorough review of the manuscript, which has significantly contributed to its improvement! Next, we will address each comment made in the review, providing a point-by-point response, in a detailed manner.
Reviewer’s remark: 1-The author needs to clarify the novelty of his work.
Author’s answer: The novelty of this research lies primarily in the approach and the focus on integrating new raw materials, such as clay and by-products from other industries on the performance of clay composites. The findings contribute with new insights to the current body of knowledge. This was further elaborated in the Introduction section of the paper.
Reviewer’s remark: 2- very little characterization technics used.
Author’s answer: Thank you for the constructive comment! We have enriched the "Materials and Methods" section with additional details. This now includes supplementary information which outlines the specific standards followed, methodologies employed, and equipment utilized in our work.
Reviewer’s remark: 3- the author should provide the industrial outcomes from this study as they used a worldwide clay as well as waste.
Author’s answer: We appreciate your valuable feedback! We updated the “Conclusions” section in the lines 614 - 620.
Reviewer’s remark: 4- the introduction is very long, from my point of view it must be shortened.
Author’s answer: We have revised the text (lines 122 – 129) to consolidate additional information into a single paragraph, thereby eliminating another one and shortening the “Introduction” chapter.
Reviewer 2 Report
- Lines 159-168: you should report the kind of instruments used for XRF and XRD and the analytical conditions. Moreover for the mineralogical analysis you should report the method for the quantitative determination;
- Line 160: you should report the origin of the clay raw material: whether it comes from a clay quarry or from a soil;
- Line 160: you should perform also the analysis about the kind of clay minerals because, as well as the amount of clay minerals, also the kind of clay mineral determine the behaviour of the clay raw material !;
- Line 162: you should report the method to obtain the grain size distribution;
- Figure 2: in the caption of the figure you should report “determined by XRD analysis (X-Ray diffraction)”;
- Table 2: in the caption, you should report “determined by XRF analysis (X-Ray Fluorescence)”
- Line 176: you should put the method for the evaluation of fineness and bulk density;
-Table 4: in the Sieve mesh sizes put dot instead of comma;
- Line 567: change “Table 10” with “Table 11”
Author Response
Dear Reviewer, we deeply appreciate your thorough review of the manuscript, which has significantly contributed to its improvement! Next, we will address each comment made in the review, providing a point-by-point response, in a detailed manner.
Reviewer’s remark: - Lines 159-168: you should report the kind of instruments used for XRF and XRD and the analytical conditions. Moreover for the mineralogical analysis you should report the method for the quantitative determination;
Author’s answer: Thank you for the constructive comment! We have enriched the "Materials and Methods" section with additional details.
Reviewer’s remark: Line 160: you should report the origin of the clay raw material: whether it comes from a clay quarry or from a soil;
Author’s answer: The necessary details have been included now in the chapter "Materials and Methods";
Reviewer’s remark: - Line 160: you should perform also the analysis about the kind of clay minerals because, as well as the amount of clay minerals, also the kind of clay mineral determine the behaviour of the clay raw material !;;
Author’s answer: Supplementary information have been included in the chapter "Materials and Methods";
Reviewer’s remark: - Line 162: you should report the method to obtain the grain size distribution;;
Author’s answer: Supplementary information have been included in the chapter "Materials and Methods";
Reviewer’s remark: - Figure 2: in the caption of the figure you should report “determined by XRD analysis (X-Ray diffraction)”;
Author’s answer: Thank you, the caption was updated with this information;
Reviewer’s remark: - Table 2: in the caption, you should report “determined by XRF analysis (X-Ray Fluorescence)” ;
Author’s answer: Thank you, the caption was updated with this information;
Reviewer’s remark: - Line 176: you should put the method for the evaluation of fineness and bulk density;
Author’s answer: Supplementary information have been included in the chapter "Materials and Methods" ;
Reviewer’s remark: -Table 4: in the Sieve mesh sizes put dot instead of comma;;
Author’s answer: Thank you, we corrected this issue;
Reviewer’s remark: - Line 567: change “Table 10” with “Table 11”;
Author’s answer: Thank you, we corrected this typo.
Reviewer 3 Report
The paper’s topic is interesting and fits the trends promoting sustainable construction materials. Building structures made of traditional materials like clay or rammed earth are perhaps not very common in modern construction. However, their share and significance have increased due to non-disputable ecological advantages. Thus, studies focused on improving the performance of such composites are welcomed.
The paper deals with the effect of various modifiers, including waste materials and by-products like limestone sludge or fly ash. The tests are adequately designed and conducted, and the conclusions are logical, justified, and practically applicable. I recommend publishing the paper; however, some imperfections must be corrected first. They are mainly editorial yet still significant.
The paper’s title is wordy, too general and does not fit with the contents. “Investigating” and “development” are an excess of words, and one is unnecessary. Moreover, the title should mention the specific topic of the paper, which is the evaluation of the impact of some modifiers on the clay plasters.
Some names of chemical substances are unnecessarily capitalized (e.g. Montmorillonite, Ozone, etc.).
Some information or statements are redundantly repeated (e.g. the paragraph started in line 122 and the next one).
It is not clear what the Authors mean by “fly” (“Mintia fly”, “third largest fly”, “ash from the fly”).
The statement that “the optimal ratio of fly ash to clay for beneficial reduction of axial shrinkage was established as 1:2” from line 324 is not justified as the shrinkage was not tested in the discussed case (fly ash was added for improving mechanical strength, not the shrinkage).
The parabola is not a good function for approximating the relationship presented in Figure 13 since the curve is monotonically rising; the hyperbole would probably be a better fit.
There are two tables named “Table 10”.
The first part of the Conclusions should be transferred into the Introduction since it justifies the significance and originality of the topic and performed tests.
With the above corrections, the paper can be published in Materials.
The text is understandable. Some minor language corrections would be appreciable.
Author Response
Dear Reviewer, we deeply appreciate your thorough review of the manuscript, which has significantly contributed to its improvement! Next, we will address each comment made in the review, providing a point-by-point response, in a detailed manner.
The paper’s topic is interesting and fits the trends promoting sustainable construction materials. Building structures made of traditional materials like clay or rammed earth are perhaps not very common in modern construction. However, their share and significance have increased due to non-disputable ecological advantages. Thus, studies focused on improving the performance of such composites are welcomed.
The paper deals with the effect of various modifiers, including waste materials and by-products like limestone sludge or fly ash. The tests are adequately designed and conducted, and the conclusions are logical, justified, and practically applicable. I recommend publishing the paper; however, some imperfections must be corrected first. They are mainly editorial yet still significant.
Reviewer’s remark: The paper’s title is wordy, too general and does not fit with the contents. “Investigating” and “development” are an excess of words, and one is unnecessary. Moreover, the title should mention the specific topic of the paper, which is the evaluation of the impact of some modifiers on the clay plasters.;
Author’s answer: Thank you for the constructive comment! We have modified the article’s title in “Development of clay composite plasters integrating industrial waste”;
Reviewer’s remark: Some names of chemical substances are unnecessarily capitalized (e.g. Montmorillonite, Ozone, etc.).;
Author’s answer: Thank you, we corrected this typo;
Reviewer’s remark: Some information or statements are redundantly repeated (e.g. the paragraph started in line 122 and the next one).;
Author’s answer: Thank you for the constructive comment, we rewritten the text (lines 122 – 129) in order to remove the redundantly information;
Reviewer’s remark: It is not clear what the Authors mean by “fly” (“Mintia fly”, “third largest fly”, “ash from the fly”);
Author’s answer: Thank you, we have corrected the text for a consistent expression and the correct use of the terminology ("fly ash");
Reviewer’s remark: The statement that “the optimal ratio of fly ash to clay for beneficial reduction of axial shrinkage was established as 1:2” from line 324 is not justified as the shrinkage was not tested in the discussed case (fly ash was added for improving mechanical strength, not the shrinkage).;
Author’s answer: Thank you for the constructive comment! We have corrected the paper.
Reviewer’s remark: The parabola is not a good function for approximating the relationship presented in Figure 13 since the curve is monotonically rising; the hyperbole would probably be a better fit.;
Author’s answer: Thank you for the constructive comment! The Figure 13 was rectified;
Reviewer’s remark: There are two tables named “Table 10”.;
Author’s answer: Thank you, we corrected this typo;
Reviewer’s remark: The first part of the Conclusions should be transferred into the Introduction since it justifies the significance and originality of the topic and performed tests.;
Author’s answer: Thank you, we considered your suggestion and made the necessary changes.